# The Controversial Roles of Areca Nut: Medicine or Toxin?

**DOI:** 10.3390/ijms24108996

**Published:** 2023-05-19

**Authors:** Pei-Feng Liu, Yung-Fu Chang

**Affiliations:** 1Department of Biomedical Science and Environmental Biology, Kaohsiung Medical University, Kaohsiung 807, Taiwan; pfliu908203@gmail.com; 2Department of Medical Research, Kaohsiung Medical University Hospital, Kaohsiung 807, Taiwan; 3Center for Cancer Research, Kaohsiung Medical University, Kaohsiung 807, Taiwan; 4Institute of Biomedical Sciences, National Sun Yat-sen University, Kaohsiung 804, Taiwan; 5Translational Research Center of Neuromuscular Diseases, Kaohsiung Medical University Hospital, Kaohsiung 807, Taiwan

**Keywords:** areca nut, component, pharmacological activity, toxicological effect

## Abstract

Areca nut (AN) is used for traditional herbal medicine and social activities in several countries. It was used as early as about A.D. 25-220 as a remedy. Traditionally, AN was applied for several medicinal functions. However, it was also reported to have toxicological effects. In this review article, we updated recent trends of research in addition to acquire new knowledge about AN. First, the history of AN usage from ancient years was described. Then, the chemical components of AN and their biological functions was compared; arecoline is an especially important compound in AN. AN extract has different effects caused by different components. Thus, the dual effects of AN with pharmacological and toxicological effects were summarized. Finally, we described perspectives, trends and challenges of AN. It will provide the insight of removing or modifying the toxic compounds of AN extractions for enhancing their pharmacological activity to treat several diseases in future applications.

## 1. The History and Usage of AN

Areca nut (AN) as the fruit of the palm tree *Areca cattechu* Linn. provides popular psychoactive substances and is used as a traditional herbal drug in countries of South and Southeast Asia, including China and India (Figure 1). AN was used for antidepression recorded in the Chinese book called YiWuZhi (The Collection of Foreign Materials) as early as the period of A.D. 25-220. AN was used as anti-parasitic medicine according to the Chinese book called YaoLu (Medicine Record) in the period of A.D. 220-265. Moreover, AN was also described to treat miasma in the Chinese book, HelinYulu, during the period of A.D. 1127-1279. In addition to its medical applications, AN also provides functions for social and religious activities [1]. AN chewers were estimated as more than 600 million people located mainly in Asian countries like India, Pakistan and Taiwan as well as migrants from South Africa and the UK, which account for at least 10% of the world population [2,3,4]. A survey from the United States showed that 17% of people self-reported that they have ever used AN products, and 31% people reported that they had friends/families of AN users, indicating the high percentage of AN usage [5]. Although AN has the potential of treating malaria, diarrhea, ascariasis, edema, stagnation of food, arthritis, and beriberi [6], it still provides a dilemma of pharmacologically useful as well as toxic effects. The biological functions of AN components and their pharmacological and toxicological effects are described in detail as below.

AN provides popular compounds with several studies focusing on it. There are several articles describing the effects caused by areca nut (AN) previously [6,7,8,9]. In addition, some of the review articles focused on specific aspects, such as carcinogenic effects [10], addiction [11], AN cessation [12], genetic and epigenetic instability [13], submucous fibrosis progression [14], oral cancer [15] and liver disease [4]. We here review the results of these reports and evaluate the controversial roles of AN.

There are more than hundred articles reporting about AN and its components which indicates the importance of AN. However, these articles covered different fields. We summarized and evaluated the recently published reports in this article to update recent trends of research in addition to acquire new knowledge about AN. Moreover, this article focused on the effect of AN (Figure 1C) but not betel nut (Figure 1D). Although AN is also called betel nut, betel nut is usually combined with other materials, such as betel leaf, betel stem inflorescence and slaked lime. In order to exclude the effect caused by the materials other than AN, we did not include reports including the additions that commonly come along with betel nut consumption.

This review contains four parts. The first part, Section 1 describes the history of AN usage. Section 2 introduces the components of AN and their biological effects. Arecoline is an important compound of AN. In addition, AN components are a mixture of different components, and AN extract has different effects from each component. The dual effects of AN with pharmacological and toxicological effects were summarized. The Section 3 introduces the pharmacological effects, and the Section 4 explores the toxicological effects of AN extract, other mixture compounds and arecoline.

## 2. AN Components and Their Biological Effects

The components of AN include alkaloids, flavonoids, tannins, triterpenoids and steroids, fatty acids and others [6,7]. Their biological activities are described below. 

### 2.1. Alkaloids

Alkaloids of AN have addictive and carcinogenic effects [16]. There are four major alkaloids, including arecoline, arecaidine, guvacoline and guvacine. Arecoline (N-methyl-1,2,5,6-tetrahydropyridine-3-carboxylic acid methyl ester) is the most abundant alkaloid and a key chemical constituent contributing to the carcinogenicity of AN [17]. The chemical structure of these components is listed in Figure 2. 

Arecaidine is the hydrolyzed product of arecoline. Arecoline and arecaidine increase collagen deposition by producing tissue inhibitors of metalloproteinases (TIMPs) in the extracellular matrix to alter the microenvironmental conditions of carcinogenic effects [18]. Guvacoline decreases exploration activity and social interaction in zebrafish [19]. Guvacine is the second abundant alkaloid followed by arecoline. The nitrosated derivate of guvacine causes DNA breaks as well as oxidative stress resulting in oral carcinogenesis [18]. Nicotine plays dual roles in anti- and pro-inflammatory effects. It exerts anti-inflammatory effects in ulcerative colitis, arthritis, sepsis, and endotoxemia but promotes diseases such as periodontitis and gingivitis due to its pro-inflammatory effects [20]. Isoguvacine, a GABA agonist, decreases blood pressure and heart rate of kidney-clipped hypertension rats by inhibiting commissural nuclei of the solitary tract [21]. 

### 2.2. Flavonoids

Flavonoids belong to polyphenolic compounds with wide biological functions for treating several disorders, such as cancer, cardiovascular complications, chronic inflammation and hypoglycemia [22]. At least 11 flavonoids were isolated from AN, including isorhamnetin, chrysoeriol, luteolin, quercetin, jacareubin and liquiritigenin [6,7]. Isorhamnetin has several pharmacological activities including cardiovascular and cerebrovascular protection, anti-inflammation, anti-tumor, antioxidation, prevention of obesity and organ protection [23]. Chrysoeriol has anticancer, antibacterial, anti-inflammatory, anti-fungal, anti-insecticide, anti-osteoporosis effects and prevents diabetes, inflammation, Parkinson’s disease, osteoporosis, and cardiovascular diseases [24]. Luteolin possesses anticancer, anti-inflammatory, antimicrobial, antioxidant and antidiabetic effects and provides health benefits for patients with Alzheimer’s disease, Parkinson’s disease, cardiac diseases and obesity [25]. Quercetin is an antioxidant, antimicrobial, antidiabetic, anticancerous and anti-inflammatory agent for preventing allergies, ulcers, microbes, cancer and Alzheimer’s disease [26]. Jacareubin is reported to have anti-inflammatory activity. The receptor-binding domain of SARS-CoV-2 Spike protein induces TNF-α synthesis and cell infiltration in the lungs of mice. Pre-treatment with jacareubin can inhibit Toll-like receptor (TLR) 4-induced lung inflammatory response caused by SARS-CoV-2 Spike protein [27]. Liquiritigenin causes antidepressant, antianxiety, antipsychostimulant, memory-enhancing and neuroprotective activities on neurodegenerative diseases, such as Parkinson’s disease, Alzheimer’s disease, stroke and brain glioma. In addition, it also has several pharmacological activities, such as radical scavenging, antibacterial and anti-inflammatory activities [28]. Another two of the new flavonoids, calquiquelignan N and calquiquelignan M, from AN provide cytotoxicity to human cancer HepG2 cell line [29].

### 2.3. Tannins

At least 12 tannins were isolated from AN, such as catechin and procyanidins [6,7]. Tannins are polyphenol antioxidants. They have anti-inflammatory, antimicrobial, antioxidant and anticancer effects and prevent cardiovascular, neuroprotective and general metabolic diseases [30]. Catechin shows effectiveness as anti-inflammatory, increasing memory, antidiabetic, anticancer, bactericidal, anti-arthritis, hepatoprotection and neuroprotection, through activation of NF-κB, TLR4/NF-κB, Nrf-2, COMT and MAPKs pathways [31,32]. Procyanidins have anti-inflammatory, antioxidant, antibacterial and anti-tumor effects. Therefore, they provide potential treatments for several diseases, such as Alzheimer’s disease, rheumatoid arthritis, diabetes, obesity, tumors and oral diseases. Among the procyanidins, procyanidin A1 inhibits the release of inflammatory factors IL-6, PGE2, NO and TNF-α by inactivating the IκB/NF-κB p65 pathway. Procyanidin A1 and B2 increase the level of antioxidant genes including HO-1, NQO1 and γ-GCS through activating the AMPK/Nrf2 pathway [33]. 

### 2.4. Terpenoids

At least eight terpenoids were isolated from AN, such as 3-carene, procurcumenol and ursolic acid [6]. Terpenoids have pharmacological effects including anti-pyretic, anti-analgesic, anti-inflammatory, antibacterial, antiviral and anticancer, such as in lung cancer [34]. 3-carene shows antibacterial activity to inhibit the growth of *Escherichia coli* in a minimum inhibitory concentration test and antioxidant activity as shown by the oxygen radical absorbance capacity activity test [35]. Procurcumenol provides anti-platelet aggregation activity in vitro promoting blood circulation [36]. In addition, triterpenoids also have biological activities including hepatoprotection, anti-inflammation, antiviral and anti-tumor [37]. Ursolic acid is a pentacyclic triterpenoid with anti-inflammatory, apoptosis induction and anticarcinogenic activities by inhibiting cancer cell proliferation and viability and preventing tumor metastatic activity and angiogenesis [38]. Ursolic acid and 3-O-acetylursolic acid arrest cell cycle and increase the apoptotic population of A375 melanoma cells by activating caspases and attenuating Bcl-2 expression [39]. 

### 2.5. Steroids

Steroids are regulators of lipid metabolism [40]. At least four steroids have been isolated from AN, such as cycloartenol and beta-sitosterol [6]. Cycloartenol increases IL-6 expression and suppresses IL-10 expression, indicating its immune potentiating effects [41]. Cycloartenol also inhibits proliferation and the colony formation by arresting Sub-G1 cell cycle and triggering apoptosis through altering the expression of Bax (Bcl-2-associated X protein) and Bcl-2 of glioma U87 cells. In addition, it inhibits the migration of glioma cells through suppressing the phosphorylation of p38 MAP kinase [42]. Cycloartenol is a putative active ingredient against type 2 diabetes mellitus through regulating the expression of AKT1, TNF-α, MAPK3 and MAPK1 pathways predicted by molecular-docking plus GO/KEGG analysis [43]. Beta-sitosterol is involved in several signaling pathways regarding cell cycle, apoptosis, proliferation, survival, invasion, angiogenesis, metastasis, anti-inflammatory, anticancer (breast, prostate, colon, lung, stomach and leukemia), hepatoprotective, antioxidant, cardioprotective and antidiabetic effects [44]. 

### 2.6. Fatty Acids

AN contains several important fatty acids, among others lauric acid, myristic acid, palmitic acid, stearic acid, arachidic acid and dodecenoic acid [6,7]. Fatty acids are key materials of anti-inflammation mediators and crucial for development, growth and preservation of human health [45]. Feeding chicks with 350 and 500 mg/kg of lauric acid monoglyceride and cinnamaldehyde improved their intestinal morphology, improved the body’s antioxidant capacity and downregulated the expression of inflammatory factors [46]. Myristic acid (200 μg/mL) can reduce skin inflammation. Myristic acid exerted anti-inflammatory activity by increasing the production of IL-10 in LPS-stimulated JA774A.1 macrophage cells. Myristic acid (12.5–100 mg/kg) also showed anti-inflammatory effects on the acute and chronic TPA-induced ear edema of mice [47]. Palmitic acid was shown to inhibit the growth and metastasis of gastric cancer. Palmitic acid (50 μM) could block the proliferation and invasion of the MGC-803 and BGC-823 gastric cells. Palmitic acid (50 mg/kg) treatment also decreased gastric cancer growth in xenografted models of nude mice. Moreover, gastric cancer cells treated with palmitic acid decreased the expression of p-STAT3, p-JAK2, N-cadherin and vimentin indicating that palmitic acid inhibited the growth of gastric cancer by blocking the STAT3 signaling pathway [48]. Stearic acid attenuates idiopathic pulmonary fibrosis by reducing profibrotic signaling. Stearic acid also reduced the expression of fibrosis markers, α-SMA and collagen type 1 and the levels of p-Smad2/3 in MRC-5 cells treated with TGF-β1. Stearic acid also reduced the levels of bleomycin-induced hydroxyproline, an amino acid present in collagen, in a mouse model indicating that stearic acid attenuates idiopathic pulmonary fibrosis through reducing the p-Smad2/3 pathway [49]. 

### 2.7. Others

The other compounds of AN also have biological effects, such as vanillin, alpha-terpineol, benzyl alcohol, capric acid, resveratrol, quinic acid, chrysophanol, physcion, p-hydroxybenzoic acid, epoxyconiferyl alcohol, protocatechuic acid, isovanillic acid, ferulic acid, vanillic acid and polyphenols. Vanillin has anti-tumor, anti-mutagenic, antioxidant, analgesic and anti-erythrocyte-sickling properties [50]. Alpha-terpineol was demonstrated to have a potential in reducing mechanical hypernociception/inflammatory response, cardiovascular and gastric lesions, anticonvulsant and antidiarrheal activities [51]. Benzyl alcohol plays important roles in bacteriostatic, anesthetic processes and relieving nerve and ganglionic pain. Benzyl alcohol also accelerates the healing process of Achilles tendon injury by increasing collagen and blood capillaries and causing fewer inflammatory cells through activating the TGF-β1/Smad2/3 pathway [52]. Capric acid has several targets including PPAR-γ, AMPA receptors, gut dysbiosis and inflammatory of oxidative stress pathways. Moreover, Capric acid exhibits an ameliorative effect of neurological diseases including affective disorders, epilepsy and Alzheimer’s disease [53]. Resveratrol shows anti-aging, anticancer and antioxidant properties [54]. Quinic acid has different biological effects, including antioxidant, aging protective, antidiabetic, anti-nociceptive, antimicrobial, anticancer, antiviral and analgesic effects. In addition, quinic acid also shows antibacterial effects by inhibiting the functions of ribosomes [55,56,57,58]. Chrysophanol has dual pharmacological and toxicity effects. Pharmacological effects include anticancer, antioxidation, neuroprotection, antibacterial, and antiviral ones and regulating blood lipids. However, the toxic effects of chrysophanol are hepatotoxic and nephrotoxic [59]. Anthraquinones are important compounds with different pharmacological effects, such as antibacterial, anti-tumor, antioxidant, laxative and other activities [60]. Physcion is a natural anthraquinone exerting anti-inflammatory, anticancer, antimicrobial and hepatoprotective properties [61]. The *p*-hydroxybenzoic acid is confirmed to have therapeutic advantages, such as antibacterial, anticancer, antioxidative and anti-inflammatory activities. However, various adverse effects including male infertility and female breast cancer were reported [62]. Epoxyconiferyl alcohol showed the cytotoxicity of human liver cancer cell lines HepG2 detected by the sulforhodamine B assay [63]. Protocatechuic acid has diverse pharmacological activities, such as antioxidant, neuroprotective, anti-inflammatory, antibacterial, anticancer, antiviral, antiosteoporotic and anti-aging activities and analgesia, protection from metabolic syndrome and preservation of kidney, liver and reproductive organs [64]. Isovanillic acid has inhibitory activity on tyrosinase and tyrosine acidase and enzymes for melanin production in mouse melanoma B16 cells. Isovanillic acid has a potential for whitening cosmetics [65]. Isovanillic acid can also attenuate fibrin polymer formation for blood clots and degrades clots by inhibiting fibrinoligase and procoagulant proteases and prolongs the time of coagulation. It is suggested that isovanillic acid has the potential as inhibitor of thrombin [66]. Ferulic acid is a phenolic compound with several pharmacological activities, especially in oxidative stress, inflammation, vascular endothelial injury, fibrosis, apoptosis and platelet aggregation [67]. Vanillic acid protects the liver by reducing transaminase levels, inflammatory cytokines and accumulation of collagen and preventing liver fibrosis. In addition, it decreases the adipose tissue of mice through the activation of the AMPK pathway [68]. Polyphenols in lipopolysaccharides of AN reduce the generation of reactive oxygen species by activating the Nrf2/HO-1 antioxidant pathways and inhibiting the MAPK pathway in RAW264.7 cells [69].

In summary, the different components of AN have their own biological or toxicological effects. Most of the components have pharmacological effects, except alkaloids and 3-acetylursolic acid have toxicological effects, and cycloartenol, chrysophanol and p-hydroxybenzoic acid have both pharmacological and toxicological effects. The biological effects of AN components are listed in Table 1. Additionally, their effects are summarized in Figure 3.

## 3. Pharmacological Effects of AN Extracts, Other Mixture Compounds and Arecoline on Several Diseases 

### 3.1. Gastric and Intestinal Diseases

Two compounds isolated from the dried fruit of AN, 2,4-dimethoxyphenyl-β-D-glucopyranoside (26) and isorhamnetin 3-O-(6”-O-α-L-rhamnopyransoyl) β-D-glucopyranoside (34), showed weak cytotoxicity and had anticancer effects against human gastric cancer cell lines (BGC-823), indicating that AN compounds have the potential for treating gastric cancer [70]. Dried pericarp of AN extracts (*Areca pericarpium* extract) and arecoline can induce the contractions of porcine lower esophageal sphincter sling and clasp muscles in a dose–response manner. However, the muscarinic receptor antagonist atropine inhibited such contractions. These results showed that AN extract and arecoline induced contractions were mediated by muscarinic receptors indicating the possibility of developing as an alternative therapy for gastroesophageal reflux disease [71]. 

### 3.2. Depression and Anxiety

The metabolites of AN analyzed with the UPLC-MS/MS system showed that 93 metabolites were capable of targeting 141 depression-related genes, including L-phenylalanine, nicotinic acid, L-tyrosine, protocatechuic acid, okanin, benzocaine, benzocaine, phloretic acid, syringic acid, cynaroside and 3,4-dihydroxybenzaldehyde. These results indicated that AN is rich in bioactive antidepression compounds [72]. Arecoline increased serotonin levels and social preference and elevated brain norepinephrine but reduced turnover of serotonin for activating central monoaminergic neurotransmission of anxiolytic effects in zebrafish through upregulation of c-fos, c-jun, egr2 and ym1. These findings suggested that arecoline possesses anxiolytic-like activity [73].

### 3.3. Liver Diseases

With AN extract treatment, hepatocellular carcinoma cells, HepJ5 and Mahlavu cells, increased the conversion of light chain 3 (LC3)-I to LC3-II, beclin levels and anti-thymocyte globulin 5+12 for ROS-mediated autophagy and lysosome formation. They also increased apoptotic proteins, such as Bax and cleaved poly (ADP-ribose) polymerase, and decreased the level of Bcl-2 for apoptosis. Tumor sizes and tumor weights in the AN extract-treated group significantly decreased compared to vehicle-treated group of nude mice. These studies demonstrated that the AN extract can inhibit the progression of hepatocellular carcinoma cells [74].

### 3.4. Neurological Disorders

Arecoline evoked an enhancement of the firing rate, including an increase of spike numbers per burst, elongation in the burst duration and the burst rate of dopaminergic neurons in the ventral tegmental area of rats. These findings may be used for potentially targeting AN cessation therapy [75]. Deficiencies of neuromuscular junctions in the motor neurons resulted in the degeneration of both neurons and muscles resulting in age-related mobility decline. Arecoline is a muscarinic acetylcholine receptor agonist capable of promoting synaptic exocytosis at neuromuscular junctions. Treatment of arecoline during the early stage of aging decreased muscle tissue and increased lifespan through the GAR-2/PLCβ pathway in the motor neurons of *Caenorhabditis elegans*. These results suggested that arecoline has a pharmacological potential to promote lifespan when it was applied at the early stage of aging [76]. Arecoline improves cognition, memory and some behavioral disorders of patients with schizophrenia or Alzheimer’s disease through activating postsynaptic muscarinic M1 receptors. Moreover, it reduced the dopaminergic hyperactivity throughout the modulation of M1, -2 and -4 receptors to ameliorate the negative symptoms of psychosis [77].

### 3.5. Bacterial Infection

Ethanolic and methanolic extracts of AN fruits have antibacterial effects against *S. aureus*, *E. aerogenes*, *E. coli* and *S. enterica* detected by both methods of minimum inhibitory concentration (MIC) and minimum bactericidal concentration (MBC). These results suggested that AN fruit extracts can serve as a natural preservative [78]. Orally administered AN extracts 14 days before the intraperitoneal challenge with *S. aureus* in rats produced increased concentrations of white blood cells but did not affect red blood cells and other cell types. In addition, no harmful effects on liver and kidney functions were observed. HPLC analysis revealed that phenolic catechin and quercetin are the major compounds of AN extracts. These findings suggested that phenolic compounds of AN extract have immunomodulatory activity against *S. aureus* infection [79]. Polyphenols of AN ethanol extraction, including catechin, epicatechin and epigallocatechin gallate, have anti-infective activities against bacteria belonging to *M. tuberculosis*, Gram-positive *S. aureus* and Gram-negative *E. coli*. These findings suggested that polyphenols of AN provide antibacterial action against multiple antibacterial resistance such as those mentioned above [80]. Several volatile components of AN fiber extracted by simultaneous hydrodistillation–extraction showed to have antimicrobial effect. Nine bacteria and yeasts were determined. The results showed that the antimicrobial effects of Gram-positive bacteria, such as *Streptococci*, were better than that of Gram-negative strains, such as *E. coli*, and the yeast *Candida albicans* [81].

### 3.6. Skin Protection

Oral administration of AN procyanidins prevented UVB-induced photoaging by detecting epidermal thickness and collagen disorientation of the dorsal skin of CD-1 mice. In addition, inhibited UVB-induced photodamage, such as expression of cyclooxygenase-2 (COX-2) and matrix metalloproteinases (MMPs), including MMP-2, MMP-9 and TIMP metallopeptidase inhibitor 1 (TIMP1), was also observed. These finding suggested that AN procyanidins could attenuate solar UVB-induced premature skin aging [82]. Overproduced melanin may cause skin disorders, such as premature aging. Tyrosinase is the critical enzyme of melanogenesis, and reduced tyrosinase activity decreased melanin production. AN extract was demonstrated to have tyrosinase inhibitory effect using 3,4-dihydroxy-L-phenylalanine (L-DOPA) as the substrate of tyrosinase [83]. Elastin is important for keeping elasticity of the skin. Elastase is a protease enzyme for the degradation of elastin. The inhibition of elastase activity can prevent skin loss of elasticity and wrinkles. AN extract was demonstrated to have elastase inhibitory effects using N-succinyl-trialanyl-paranitroanilide as the substrate of elastase [83]. Taken together, AN extract has an anti- aging skin potential.

### 3.7. Anthelmintic Activity

AN crude aqueous extract damaged the morphology and average number of eggs of the nematode, *Ascaridia galli*, found in chicken. The body weight, erythrocyte, leukocyte and hemoglobin were increased in chicken with AN extract treatment. These data indicated that AN extract has anthelmintic efficacy against *A. galli* in chicken [84]. AN extract was found to change the morphology and decrease the motility time of flukes *Fasciola* spp., from the liver of buffaloes. This demonstrates that AN extract has anthelmintic efficacy against liver fluke [85].

### 3.8. Antioxidant

The DPPH (2,2-diphenyl-1-picryl-hydrazyl-hydrate) assay is used to detect the scavenging ability on stable DPPH radicals of compounds with hydrogen-donating capacity. A polysaccharide PAP1b isolated from AN containing mannose, galactose, xylose and arabinose showed the ability of scavenging hydroxyl radicals using a DPPH assay in vitro. Therefore, this polysaccharide from AM had a potential antioxidant effect for the food and pharmaceutical industry [86]. The ABTS (2,2′-azino-bis (3-ethylbenzothiazoline-6-sulfonic acid)) assay is used to detect the ability of transferring hydrogen atoms for neutralizing ABTS radical cations of compounds. AN ethanol extraction with phenolic content showed high free radical scavenging activity detected by both DPPH and ABTS assays. It suggested that the phenolics of AN have high antioxidant capacities [83]. 

### 3.9. Anti-Hypoxia

AN polyphenol has antioxidant, anti-inflammatory and antibacterial effects. Recent studies showed that they have protective effects on oxidative stress caused by hypoxia via improving blood gas index of hypoxic organism through scavenging excessive free radicals. These findings indicated that AN polyphenol can be used as potential anti-hypoxia drugs [87]. 

### 3.10. Anti-Osteoarthritis

Ethanol extracts of AN show anti-inflammatory effects on lipopolysaccharide (LPS)-induced inflammation through decreasing NO and expressing materials, inducible nitric oxide synthase (iNOS) and cyclooxygenase-2 (COX-2). It also suppressed carrageenan-induced inflammation in rats by decreasing edema and prostaglandin E2 levels. Therefore, AN has a potential as an anti-osteoarthritis agent [88].

In summary, AN extracts, other mixture compounds and arecoline have several pharmacological effects to prevent certain diseases that are listed in Table 2.

## 4. Toxicological Effects of AN Extract, Other Mixture Compounds and Arecoline on Several Diseases 

### 4.1. Oral Diseases

AN chewing is supposed to be the major etiological factor in the manifestation of oral submucous fibrosis through dysregulating of proinflammatory cytokines, such as transforming growth factor-β [89]. However, a systematic meta-analysis showed that the prevalence of oral submucous fibrosis in AN chewers was 5% suggesting a low prevalence of oral submucous fibrosis in AN chewers. These results indicate that ANs may not be the sole cause of oral submucous fibrosis [90]. However, AN chewing is significantly associated with a negative prognosis for patients with oral cancer [91]. AN extract decreased viability of human buccal epithelial KB cells. It promotes apoptosis by increasing Bax and decreasing BCL2. The AN extract also elevated cell death by decreasing cell cycle regulators, such as cyclin D1, cyclin E1, cyclin dependent kinase 4 (CDK4), retinoblastoma (Rb), proliferation cell nuclear antigen (PCNA) and tumor suppressor p53, through increasing transcription factors, including Activator Protein -1 (AP-1) subunits (c-Jun/c-Fos), p21 and p16. AP-1 is a transcription factor involved in oncogenesis. These findings suggested that AN regulated AP-1 signaling of KB cells for oral carcinogenesis [92]. Arecoline is the major compound in AN to initiate the oral submucous fibrosis process [93]. It can induce reactive oxygen species, cell cycle arrest, apoptosis and DNA damage [94]. Arecoline-induced oral pathologies are explained by the following mechanisms: a suppression of p53 for DNA repair and increased reactive oxygen species (ROS) for epithelial-mesenchymal transition (EMT) and metastasis through mitogen-activated protein kinase (MAPK), phosphatidylinositol-3-kinase (PI3K)/Akt and nuclear factor kappa-light chain enhancer of activated B-cells (NF-κB) and PKC–protein kinase C (PKC) pathways [95]. It alters the expression of several inflammatory cytokines, such as serum amyloid A1, interleukin-6, IL-36G, chemokine CCL2 and CCL20, for epithelial-mesenchymal transformation (EMT) in oral squamous cell carcinoma cells. Moreover, arecoline enhanced cervical lymph node metastasis of tongue xenografted models using nude mice. These results also suggested that arecoline induces EMT and promotes metastasis of oral cancer [96]. The phosphodiesterase (PDE)/cAMP/Epac/C-EBP-β signal cascade regulates renal fibrosis in renal tubular epithelial cells [97]. Transforming growth factor-β (TGF-β) exerts pathological effects on organ fibrosis. Arecoline enhanced phosphodiesterase 4A activity. Silence phosphodiesterase 4A reversed the effects of TGF-β-induced fibrotic changes caused by arecoline. These findings indicated that arecoline promoted TGF-β1-induced buccal mucosal fibroblast activation through enhancing phosphodiesterase 4A activity during oral submucosal fibrosis [98]. Oxidative stress-caused DNA damage also leads to nuclear anomalies such as micronuclei. The levels of glutathione antioxidant enzymes, such as reductase and superoxide dismutase, were reduced, but the micronuclei count increased in buccal exfoliated cells of AN chewers. The detection of micronuclei in AN chewers can be a useful risk biomarker for oral cancer diagnosis [99]. However, there is no direct evidence of arecoline-induced carcinogenesis in animal models, possibly because of inappropriate treatment procedure or being caused by the biological differences between human and mice or rats [100]. 

### 4.2. Liver Diseases

Twelve of fifteen studies indicated that AN is a risk factor for various liver diseases like liver cirrhosis, hepatocellular carcinoma, complicating cirrhosis, liver fibrosis and chronic liver disease [4]. Most of these studies came from Taiwan, suggesting a high prevalence and many adverse effects of AN on liver in Taiwan. Moreover, five studies showed an increased risk of hepatocellular carcinoma and liver cirrhosis associated with an increased consumption of betel quid. Arecoline increased the proliferation, increased the migration ability of the HepG2 cells and upregulated the expression of PI3K-AKT pathway factors. These findings suggested that arecoline promoted migration and proliferation of human HepG2 cells by activating the PI3K/AKT/mTOR pathway [101]. Metabolic syndrome is related to liver fibrosis, cirrhosis and hepatocellular carcinoma. Patients of both metabolic syndrome and betel quid chewing had a higher risk of liver fibrosis than those with neither metabolic syndrome nor betel quid chewing. Betel quid chewing was reared with liver fibrosis in patients with metabolic syndrome but not in those of without metabolic syndrome [102]. In addition, betel nut chewing was associated with a high risk of liver fibrosis in subjects with nonalcoholic fatty liver disease [103]. 

### 4.3. Behavior and Addiction

AN alkaloid with psychoactive activity, including arecoline, arecaidine, guvacine and guvacoline, was reported to cause behavioral alterations in zebrafish [19]. They were reported to change behaviors, such as increasing the sense of well-being, stamina and euphoria. A review article described that AN chewing claimed to produce warm sensations of the human body, palpitation and heightened alertness and tolerance to hunger. All these neurological effects may be caused by the chemicals in AN that affect the autonomic nervous system at various levels [104]. Injected dichloromethane of AN into rat brain has antidepressant properties via monoamine oxidase-A inhibition [105]. Addiction is a significant public health menace. AN cessation is needed for the chewers. A total of 16 studies showed that the determinants influence the results of AN cessation including addiction of AN, withdrawal symptoms and sociocultural factors [106]. The majority of intervention method for AN cessation is behavior control. Behavior changing interventions were proven to be more effective [104]. However, pharmacological cessation therapies are also emerging for controlling this public health problem [11]. The results of a literature search related to betel nut use provided 139 references between the years 1970 and 2019 demonstrating a lack of up-to-date statistics on betel nut use. Therefore, appropriate policies, educational and cessation programs of helping to control betel nut applications are warranted [107].

### 4.4. Cardiac Diseases

Intraperitoneally injected arecoline in rats induced cardiac apoptosis through the Fas/Fas ligand pathway [108]. In addition, arecoline causes of cardiotoxicity and heart damage through pathways, such as induced IL-6-activated JAK2/STAT3, MEK5/ERK5 and mitogen-activated protein kinases (MAPK) pathways [109]. Intraperitoneally injected arecoline also caused cardiac fibrosis in rats through transforming growth factor-β (TGF-β)/Smad-mediated signaling pathways [110]. 

### 4.5. Gastric and Intestinal Diseases

Feeding raw AN extract developed gastric cancer in mice. Carcinogenesis was induced here by histone H3 epigenetic modifications and the Rb/E2F1 pathway [111]. The constituents of intestinal epithelial cells, such as total hexose, sialic acid, cholesterol and enzymes, including alkaline phosphatase, Ca2^+^-Mg^2+^-ATPase and sucrase, were decreased in rat administrated with AN extract by gastric intubation. These findings suggested that prolonged chewing of AN decreased intestinal epithelial cell lining functions [112].

### 4.6. Genotoxicity 

Arecoline and its metabolites were reviewed to have genetic toxicology, such as gene mutations, DNA damage and repair endpoints as well as cytogenetic effects by in vitro and in vivo experiments [17]. 

### 4.7. Reproduction and Development

Eating AN before pregnancy was followed by a higher preterm birth rate in China [113]. The results of a systematic database review containing 15,270 women of 28 studies showed that prenatal betel nut use is associated with a low birth weight [114]. Arecoline caused oocyte meiosis arrest by disrupting spindle assembly, actin filament dynamics and kinetochore-microtubule attachment stability in mouse oocytes. In addition, arecoline reduced ATP level and increased oxidative stress by disturbing the distribution of mitochondria to induce oocyte apoptosis [115]. Myogenic differentiation is an important process during embryonic development. The inhibitory effects of myogenesis through the signal transducer and activator of transcription 3 (Stat3) pathway caused by arecoline was reviewed [116]. Fortunately, the arecoline induced defective effects of myogenesis, including decreased nuclei in each myotube and the number of myotubes, and the level of the myogenic markers, such as myosin heavy chain and myogenin, in C2C12 myoblast cells can be prevented by N-acetyl cysteine through an extracellular signal-regulated kinase (ERK) pathway [117].

### 4.8. Kidney Disease

The analysis of 43,636 men from Taiwan Biobank showed that a high risk of kidney stone disease was found in subjects with betel nut chewing habits. Moreover, those who chewed more than 30 quids betel nut daily have more than 1.5-fold possibility to get a kidney stone [118]. Arecoline increased cell migration and the expression of epithelial mesenchymal transition and fibrogenesis markers, such as N-cadherin, vimentin, α-SMA and collagen but decreased another epithelial mesenchymal transition marker, E-cadherin, in human kidney (HK2) cells. The above results demonstrated that arecoline played an important role in inducing EMT and fibrogenesis of renal tubule cells for promoting the progression of chronic kidney disease [119]. 

### 4.9. Neuron Activation

Arecoline induced activation of nicotinic acetylcholine receptors (nAChR) in *Xenopus* oocytes expressing either human α4 and β2 nAChR subunits or α7 subunits. Arecoline analogs, isoarecolone and 1-(4-methylpiperazin-1-yl) ethanone, had the α4 nAChR selectivity without muscarinic activity. These finding indicated that α4 nAChR is at the root of nicotine addiction for betel nut addiction [120].

### 4.10. Abortifacient

AN was reported to have an abortifacient effect in rats and is one of the herbs considered unsafe for use during pregnancy [121].

In summary of Section 4, AN extracts, other mixture compounds and arecoline have several toxicological effects that induce or promote different diseases. Diseases associated with toxicological effects of AN are listed in Table 3.

Overall, both pharmacological and toxicological effects of AN extracts, additions of combined compounds and arecoline are presented in Figure 4. Both males and a females have the same effects, except reproduction. This figure does not indicate any gender. This figure is hand-drawn using Inkscape (https://inkscape.org/zh-hant/, accessed on 8 May 2023 to 10 May 2023).

## 5. Perspectives

By comparing the effects of AN components, we found that most of the components are useful in medicinal applications for a diverse array of ailments. However, there are also some toxic compounds, such as arecoline, arecaidine and guvacine, that induce toxicological effects related to carcinogenicity, increased DNA breaks and oxidative stress. In the future, removing or modifying the toxic ingredients of AN extractions may improve their pharmacological effects, which may effectively and safely treat several diseases such as cancer or speed up drug development.

One of the purposes of this review was to find out the trends and challenges of AN from recent published reports. We found several trends and perspectives. (1) One of the common functions of AN in Traditional Chinese Medicine is its anthelmintic function. However, research on anthelmintic efficacy for human was cautioned recently, due to concerns about its accompanying toxicity. However, the anthelmintic efficacy on other animals, such as chicken and buffaloes [84,85] was promising, suggesting that the toxicity of AN may also be tolerated by us. (2) Several articles studied the behavior and addiction/cessation [11,19,104,105,106], indicating one of the trends of AN research. Since arecoline activates muscarin M1 receptors, reports about the effect of behavior and addiction by AN [75,76,77] provide accelerating evidence. (3) Most articles were studying oral diseases [89,90,91,92,93,94,95,96,98,99,100]. The effects of AN on other organs, such as the liver, heart and kidney, and cells such as myoblasts [116] were recently published, indicating more diverse effects of AN than hitherto expected. (4) Several toxic materials also serve as medicine, indicating the potential of toxins as medicines, such as arecoline. Arecoline is a well-known toxin. It causes anxiolytic behavior in zebrafish [73]. To remove its toxicity by modifying its structure may be applied to animal models and hopefully to humans in the near future. The structures of four areca alkaloids are similar but have a different extent of anxiolytic-like effects in zebrafish. Therefore, the structural modification of toxic materials from AN by chemistry may develop new potential for clinical applications. (5) Volatile components of AN also provide antimicrobial effects [81]. Volatile compounds have the potential for development in fragrances for cosmetic, pharmaceutical or clinical applications. (6) Social interaction causes the popular usage of AN. It will be a great challenge to change the public opinion about the toxicological risks of AN usage. Therefore, information and education about the harmful effects of AN consumption are warranted. 

## 6. Conclusions

AN is used for traditional herbal medicine and social activities in several countries. It was used as early as about the years A.D. 25-220 in China. The traditional usage of AN was applied for treating several diseases including malaria, diarrhea, ascariasis, edema, stagnation of food, arthritis and beriberi. However, it was also reported to have toxicological effects. Knowing the effects of single compounds as well as their synergistic interactions will be helpful for realizing the functions of AN. Effective components of AN include among others alkaloids, flavonoids, tannins, triterpenoids, fatty acids and steroids. The pharmacological effects of AN include curing gastric and intestinal diseases, depression and anxiety, liver diseases, nerve diseases, bacterial infections, skin protection, anthelmintic efficacy, antioxidant, anti-hypoxia and anti-osteoarthritis. The toxicological effects of AN include oral diseases, liver diseases, behavior and addiction, cardiac diseases, gastric and intestinal diseases, genotoxicity, reproduction and development, kidney disease, neuron and abortifacient. This article summarizes components of AN and their pharmacological and toxicological effects in several diseases. In the future, removing or modifying the toxic ingredients of AN extractions may improve their pharmacological effects, which may effectively and safely treat several diseases such as cancer cells or speed up the development of drugs. Moreover, AN extracts may be applied in veterinary medicine such as poultry as an anthelmintic agent, and the volatile components may be developed as fragrances. The effects of AN on organs and the oral cavity are emerging. Further studies will explore the effects of AN on animals. A better understanding of the mechanism of arecoline on the activation of muscarinic M1 receptors may provide mechanisms of action for behavior and addiction problems caused by AN. Finally, to modify the structure of toxic components by synthetic chemistry will be helpful for many of the above purposes.

## Figures and Tables

**Figure 1 ijms-24-08996-f001:**
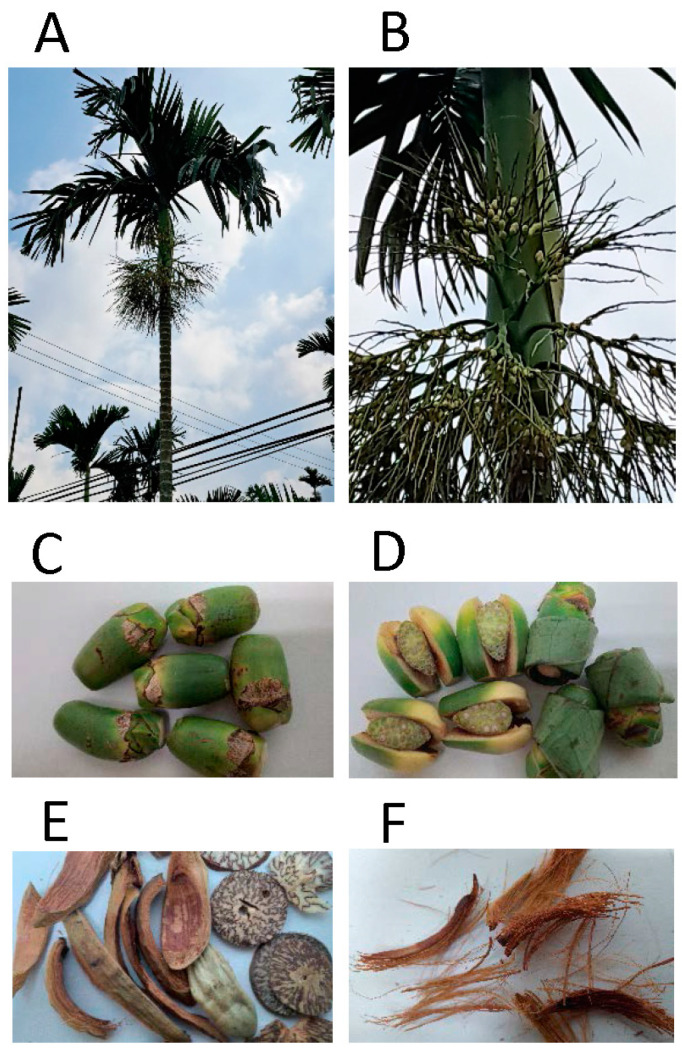
The *Areca cattechu Linn.* palm tree and its fruits (AN). (**A**) Habitus of the *Areca cattechu Linn.* palm tree. (**B**) Fruit stands of the *A. cattechu* tree. (**C**) Areca nuts with tip removed are used for chewing. (**D**) AN alone or combined with betel leaf, betel stem inflorescence and slaked lime is a popular psychoactive drug in subtropical and tropical countries. (**E**) Dried AN boiled in water is applied in traditional Chinese medicine. (**F**) Fibers of ANs provide substantial risks for oral diseases.

**Figure 2 ijms-24-08996-f002:**
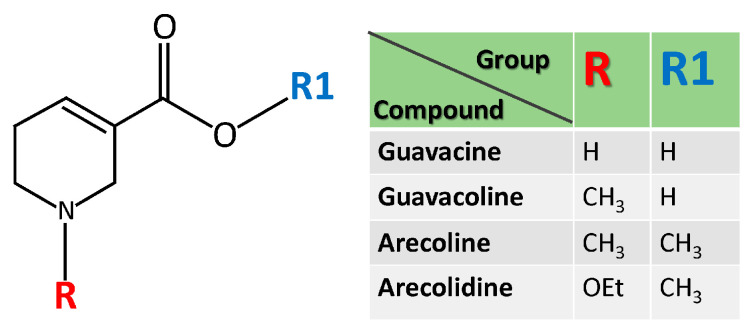
The chemical structure of areca alkaloids includes arecoline, arecaidine, guvacoline and guvacine. Their structures are similar but carry different functional groups, R1 and R2.

**Figure 3 ijms-24-08996-f003:**
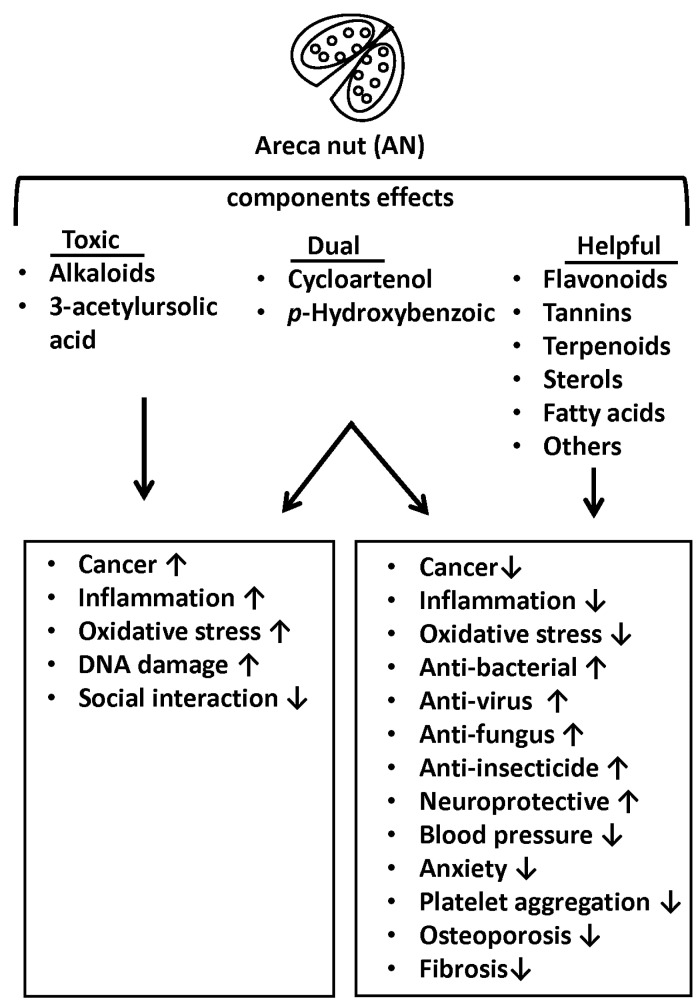
Pharmacological and toxicological effects of AN components. Most of the AN components are helpful for organisms from a health point of view. However, some AN components have toxicological effects. ↑: Activated; ↓: Inhibited.

**Figure 4 ijms-24-08996-f004:**
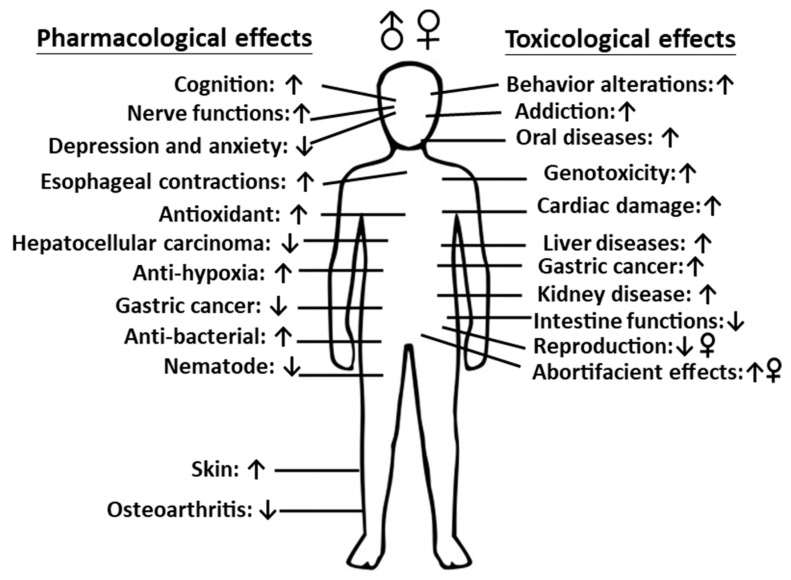
The distribution of pharmacological and toxicological effects of AN extract, combined compounds and arecoline in the human body. The effects in other animals, such as mice, rats, porcine, zebrafish, *C. elegans*, chickens and buffaloes were included from different sources cited in this review. ↑: Activated, ↓: Inhibited.

**Table 1 ijms-24-08996-t001:** The biological effects of AN components.

Effects	Components	Biological Materials	References
**Cancer**	**↑**: **Alkaloids**		[16]
	Arecoline	Hep-2 and KB cells	[17]
	Arecaidine	fibroblasts	[18]
	Guvacine	human buccal epithelial cells	[18]
	**↓**: **Flavonoids**		
	Isorhamnetin	>8 cells of different cancer types, mice	[23]
	Chrysoeriol	>3 cells of different cancer types, mice	[24]
	Luteolin	>9 cells of different cancer types	[25]
	Quercetin	>5 cells of different cancer types, mice	[26]
	**↓**: **Tannins**		
	Catechin	HCT-116 cells	[31]
	Procyanidins	>4 cells of different cancer types	[33]
	**↓**: **Terpenoids**		
	Ursolic acid	mice, human	[38]
	**↓**: **Steroids**		
	Cycloartenol	U87 cells	[42]
	β-sitosterol	>16 cells of different cancer types, rat	[44]
	**↓**: **Fatty acids**		
	Palmitic acid	MGC-803, BGC-823 cells, mice	[48]
	**↓**: **Others**		
	Vanillin	cells, rats	[50]
	Resveratrol	breast cancer cells, A549, HeLa, DU-145, HepG2	[54]
	Quinic acid	human	[55]
	Chrysophanol	HepG2, MCF-7, T47D, HCT116, A549 cells	[59]
	Anthraquinones	>20 cells of different cancer types, mice	[60]
	Physcion	in vitro, in vivo	[61]
	p-Hydroxybenzoic acid	rats	[62]
	Epoxyconiferyl alcohol	HepG2	[63]
	Protocatechuic acid	>6 cells of different cancer types	[64]
**Inflammation**	**↑**: **Steroids**		
	Cycloartenol	human blood	[41]
	**↓**: **Alkaloids**		
	Nicotine	human	[20]
	**↓**: **Flavonoids**		
	Isorhamnetin	>6 cells, mice, rats	[23]
	Chrysoeriol	RAW264.7, HaCaT cells, mice, rats	[24]
	Luteolin	264.7, HUVEC cells, mice, rats	[25]
	Quercetin	RAW264.7, HUVEC cells, rats, human	[26]
	Jacareubin	mice	[27]
	Liquiritigenin	RAW264.7 cells, rats, mice	[28]
	**↓**: **Tannins**		
	Catechin	human stomach cancer cell lines	[31]
	Procyanidins	RAW264.7 cells, rats, mice	[33]
	**↓**: **Terpenoids**		
	Ursolic acid	mice, rats	[38]
	**↓**: **Steroids**		
	β-sitosterol	J774A.1 cells, mice, rats	[44]
	**↓**: **Fatty acids**		
	Lauric acid	chicks	[46]
	Myristic acid	JA774A.1 cells, mice	[47]
	**↓**: **Others**		
	α-Terpineol	mice	[51]
	Benzyl alcohol	rats	[52]
	Capric acid	IPEC-J2, COS-7 cells	[53]
	Physcion	MH7A cells	[61]
	p-Hydroxybenzoic acid	rats	[62]
	Protocatechuic acid	PC12, CGNs 2. BV2, cells, mice, porcine	[64]
	Ferulic acid	mice, rat	[67]
	Vanillic acid	mice,	[68]
**Oxidative stress**	**↑**: **Alkaloids**		
	Guvacine	human	[18]
	**↓**: **Flavonoids**		
	Isorhamnetin	RPE, H9C2, C2C12 cells	[23]
	Luteolin	HT-29, SNU-407 cells, rats	[25]
	Quercetin	SH-SY5Y cells, rats	[26]
	Liquiritigenin		[28]
	Calquiquelignan N and M	HepG2 cells	[29]
	**↓**: **Tannins**		
	Procyanidins	RAW264.7 cells	[33]
	**↓**: **Steroids**		
	β-sitosterol	RAW 264.7, HT22 cells	[44]
	**↓**: **Others**		
	Vanillin	B16F0 cells	[50]
	Capric acid	SY5Y, Neuro2a, AML12 cells	[53]
	Resveratrol	none	[54]
	Quinic acid	none	[57]
	Chrysophanol	BV2 cells, mice	[59]
	Anthraquinones	rats	[60]
	p-Hydroxybenzoic acid	CGN cells	[62]
	Protocatechuic acid	HUVEC cells, rats	[64]
	Ferulic acid	HEK293, SH-SY5Y, HUVEC cells, rats	[67]
	Polyphenols	RAW264.7 cells	[69]
**Antibacterial**	**↑**: **Flavonoids**		
	Chrysoeriol	>5 Gram positive bacteira, >5 Gram negative bacteira	[24]
	Quercetin	*S. aureus*, *S. saprophyticus*, *E. coli*, *P. aeruginosa*	[26]
	Liquiritigenin	*M. tuberculosis*, *M. bovis*, *S. aureus*, *S. epidermidis*, *S. hemolyticus*	[28]
	**↑**: **Tannins**		
	Catechin	*S. aureus* and *L. monocytogenes*	[31]
	Procyanidins	*S. mutans*, *B. cereus*	[33]
	**↑**: **Terpenoids**		
	3-carene	*E. coli*	[35]
	**↑**: **Others**		
	Benzyl alcohol	rats	[52]
	Quinic acid	*P. aeruginosa*	[56]
	Anthraquinones	*S. aureus*, *S. epidermidis*, *P. aeruginosa*, *E. faecalis*, *P. vulgaris*, *P. mirabilis*, *S. typhi*, *E. cloacae*, *E. aerogenes*, *K. pneumoniae*, *Vibrio harveyi*	[60]
	Physcion	*S. aureus*, *P. aeruginosa*, *C. albican*, *M. gypseum*	[61]
	p-Hydroxybenzoic acid	rats	[62]
	Protocatechuic acid	*E. coli*, *P. ceruminous*, *S. aureus*, *B. cereus*, *S. pneumoniae*, *A. barramundi*, *H. pylori*	[64]
**Antivirus**	**↑**: **Others**		
	Quinic acid	RNA-dependent RNA polymerase (RdRp) of the SARS-CoV-2	[58]
	Chrysophanol	JEV virus	[59]
	Protocatechuic acid	HBV, NDV,	[64]
**Anti-fungus**	**↑**: **Flavonoids**		
	Chrysoeriol	*F. graminearum*, *P. graminicola*	[24]
**Anti-insecticide**	**↑**: **Flavonoids**		
	Chrysoeriol	*A. pisum*, *R. meliloti*, *S. litura*	[24]
**Neuroprotective**	**↑**: **Flavonoids**		
	Liquiritigenin	D10 cells	[28]
	**↑**: **Tannins**		
	Catechin	rats	[32]
**DNA damage**	**↑**: **Alkaloids**		
	Guvacine	human buccal epithelial cells	[18]
**Blood pressure**	**↓**: **Alkaloids**		
	Isoguvacine	rats	[21]
**Social interaction**	**↓**: **Alkaloids**		
	Guvacine	zebrafish	[19]
**Anxiety**	**↓**: **Flavonoids**		
	Liquiritigenin	mice	[28]
**Platelet aggregation**	**↓**: **Terpenoids**		
	Procurcumenol	rats	[36]
**Osteoporosis**	**↓**: **Flavonoids**		
	Chrysoeriol	MC3T3-E1 cells	[24]
**Fibrosis**	**↓**: **Fatty acids**		
	Stearic acid	MRC-5 cells, mice	[49]

↑: Activated; ↓: Inhibited.

**Table 2 ijms-24-08996-t002:** Pharmacological effects and mechanisms of AN extract, other mixture compounds and arecoline for several diseases.

Diseases	Pharmacological Effects	AN Components	Mechanism	Biological Materials	References
**Gastric and intestine**	against gastric cancer cell	15.91 μM 2,4-Compound 26, 20.13 μM Compound 34		BGC-823 cells	[70]
	esophageal sphincter muscle contractions ↑	300 ng/L AN pericarp extracts, 300 nM arecoline		porcine	[71]
**Depression and anxiety**	antidepression		141 depression-related genes		[72]
	anxiolytic-like activity	10 mg/L arecoline	↑ c-fos, c-jun, egr2 and ym1	zebrafish	[73]
**Liver**	hepatocellular carcinoma progression ↓	20~30 µg/mL or 20 mg/kg AN extract	↑ autophagy and apoptosis	HepJ5 and Mahlavu cells, nude mice	[74]
**Nerve**	dopaminergic neuron firing rate ↑	0.2 mg/kg arecoline	↑ dopaminergic	rats	[75]
	synaptic exocytosis at neuromuscular junctions ↑	0.2 mM arecoline	↑ GAR-2/PLCβ pathway	*C. elegans.*	[76]
	improving cognition, memory and behavioral disorders	arecoline	↑ muscarinic M1 receptors	human	[77]
**Bacterial infection**	antibacterial effects	1–100 mg/mL AN extracts		*S. aureus*, *E. aerogenes*, *E. coli*, *S. enterica*	[78]
	against *Staphylococcus aureus*	1000 mg/kg BW AN extracts		*S. aureus*, rats	[79]
	against *Mycobacterium tuberculosis*, *Staphylococcus aureus*, *Escherichia coli*	0.975 ± 0.02 μg/mL AN extracts		*M. tuberculosis*, *S. aureus*, *E. coli*.	[80]
	antimicrobial activity of nine Gram-negative, Gram-positive bacteria and yeast	10.3 ± 1.1–40.0 ± 3.0 AN SHDE extract		*B. subtilis*, *E. faecalis*, *E. coli*, *P. aeruginosa*, *S. aureus*, *S. agalactiae*, *S. canis*, *S. pyogenes*, *Candida albicans*	[81]
**Skin protection**	UVB-induced photoaging ↓	AN procyanidins	↓ COX-2, MMPs	mice	[82]
	tyrosinase and elastase inhibition	295.79 ± 11.97 mg/g dry weight AN extract	↓ Tyrosinase, elastase		[83]
**Anthelmintic activity**	against the nematode *Ascaridia galli*	250 mg/mL, 100 mg/mL AN extract		*Ascaridia galli*, chicken	[84]
	against the fluke *Fasciola* spp.	10%, 20%, 40% AN extract		*Fasciola* spp., buffaloes	[85]
**Antioxidant**	antioxidant	2 mg/mL PAP1b	↓ hydroxyl radicals		[86]
	antioxidant	20 µL AN extract	↓ free radical		[83]
**Anti-hypoxia**	protecting effects caused by hypoxia	AN polyphenols	↓ free radicals		[87]
**Anti-osteoarthritis**	suppressed inflammation	1 and 10 mg/kg/day AN ethanol extract	↓ iNOS, COX-2	rats	[88]

↑: Activated, ↓: Inhibited.

**Table 3 ijms-24-08996-t003:** The toxicological effects and mechanisms of AN extract, other mixture compounds and arecoline resulting in several diseases.

Diseases	Toxicological Effects	AN Components	Mechanism	Biological Materials	References
**Oral**	dysregulating proinflammatory cytokines	AN	↓ TGF-β	human	[89]
	low prevalence of oral submucous fibrosis			human	[90]
	deteriorate prognosis of oral cancer	AN		human	[91]
	oral carcinogenesis	0.35% AN extract	↑ AP-1, apoptotic gnes, ↓ cell cycle regulators	KB cells	[92]
	initiate oral submucous fibrosis process	Arecoline		human	[93]
	reactive oxygen species ↑, cell cycle arrest, apoptosis ↓, DNA damage	800 μg/mL Arecoline		OSCC cells	[94]
	induce oral pathologies mechanisms	Arecoline	↑ MAPK, PI3K Akt, NF-κB, PKC pathways		[95]
	promote metastasis of oral cancer	160 μg/mL Arecoline	cytokines for EMT	CAL33 and UM2 cells	[96]
	promoted buccal mucosal fibroblasts	20 and 50 µg/mL Arecoline	↑ TGF-β	buccal mucosal fibroblasts	[98]
	increased oxidative stress	AN	↓ reductase, superoxide and dismutase, ↑ micronuclei in buccal exfoliated cells	human	[99]
	carcinogenesis not found in animal models	Arecoline			[100]
**Liver**	risk factor for various liver diseases	AN		human	[4]
	promoting migration and proliferation of human HepG2 cells	2.5 µM Arecoline	↑ PI3K-AKT pathway	HepG2 cells	[101]
	associated with liver fibrosis	AN		human	[102]
	high risk of liver fibrosis	AN		human	[103]
**Behavior and Addiction**	behavioral alterations	0.001, 0.01, 0.1, or 1 ppm four alkaloid		zebrafish	[19]
	increase sense of well-being, stamina, and euphoria	AN		human	[104]
	antidepressant	10 mg/kg Dichloromethane fraction	↓ monoamine oxidase-A	rat brain	[105]
	addiction of AN	AN		human	[106]
	methods for AN cessation	AN		human	[104]
	emerging pharmacological cessation therapies	AN		human	[11]
**Cardiac**	cardiac apoptosis	5 and 50 mg/kg/day arecoline	↑ Fas/Fas ligand pathway	rats	[108]
	cardiotoxicity and heart damage	5 and 50 mg/kg/day arecoline	↑ JAK2/STAT3, MEK5/ERK5, MAPK pathways	rats	[109]
	cardiac fibrosis	5 and 50 mg/kg/day arecoline	↑ TGF-β)/Smad pathways	rats	[110]
**Gastric and intestinal**	developed gastric cancer	1 mg raw areca nutt	↑ histone H3 epigenetic modifications and Rb/E2F1 pathway	mice	[111]
	decreases intestinal epithelial cell lining functions	AN extract	↓ membrane constituents	rats	[112]
**Genotoxicity**	genetic toxicology	Arecoline	↑ gene mutations, DNA damage and repair, cytogenetic effects	in vitro, in vivo	[17]
**Reproduction and development**	risk factor for preterm birth	AN	↑ oxidative stress	human	[113]
	low birth weight	AN		human	[114]
	inducing oocyte apoptosis	160, 180 and 200 μg/mL arecoline	↑ oxidative stress	mouse oocytes	[115]
	Inhibiting myogenesis	0.04 and 0.08 mM arecoline	↓ pStat3	C2C12 cells	[116]
	defective effects of myogenesis	0.04 and 0.08 mM arecoline	↓ myosin heavy chain, myogenin	C2C12 cells	[117]
**Kidney**	high risk of kidney stone disease	>30 AN quids		human	[118]
	promoting the progression of chronic kidney disease	10, 20, or 40 μg/mL arecoline	↑ EMT genes, fibrogenesis markers	HK2 cells	[119]
**Neuron activation**	nicotine addiction	100 µM four areca alkaloid	↑ nAChR	*Xenopus* oocytes	[120]
**Abortifacient**	may have abortifacient effects	AN		human	[121]

↑: Activated, ↓: Inhibited, AN: partial or whole AN chewing.

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
