# Peer review of "The Controversial Roles of Areca Nut: Medicine or Toxin?"

_ijms, 2023, doi:10.3390/ijms24108996_

Round 1

Reviewer 1 Report

In current review authors have compiled literature available on pros and cons for areca nut use. However, manuscript lack novelty. Recent literature is available e.g., DOI: 10.4103/0971-5851.133702; https://doi.org/10.1007/s00204-020-02926-9; 10.5005/jp-journals-10031-1041; https://doi.org/10.1002/fft2.70.

2. Authors reported that "Areca alkaloids of AN have addictive and carcinogenic effects " however at what dose its showed carcinogenic effects, it is not mentioned.

3. Authors also reported so many beneficial pharmacological effects of various class of compounds of areca nut and again the at what dose they showed these effects are not given. 

4. Chemical structures of compounds should be given.

-----------------------

NA

Author Response

In current review the authors have compiled literature available on pros and cons for areca nut use. However, manuscript lack novelty.

Ans: Our revised version was updating the newest publications and highlighting the comparison of two dual effects of areca nut and of their compounds as a novel attempt.

Recent literature is available e.g., DOI: 10.4103/0971-5851.133702; https://doi.org/10.1007/s00204-020-02926-9; 10.5005/jp-journals-10031-1041; https://doi.org/10.1002/fft2.70. Garg, A.; Chaturvedi, P.; Gupta, P. C. A review of the systemic adverse effects of areca nut or betel nut. Indian J Med Paediatr Oncol 2014, 35, 3-9

Ans: Thanks for the suggestion. We have included this article suggested by the reviewer in our revised manuscript as reference 8. We particularly focused on the literature of the recent 5 years. That is why we did not cite above article in the unrevised version of the MS.

  1. Authors reported that "Areca alkaloids of AN have addictive and carcinogenic effects " However at what dose it showed carcinogenic effects, it is not mentioned.

Ans: The doses were added in Table 3.

  1. Authors also reported so many beneficial pharmacological effects of various class of compounds of areca nut and again at what dose they showed these effects are not given. 

Ans: The doses were added in Table 2.

  1. Chemical structures of compounds should be given.

Ans: The chemical structures of alkaloids, the most important molecules of AN, were added in Figure 2.

Reviewer 2 Report

ACCOUNT

The review article “The controversial roles of areca nut: medicine or toxin” by Pei-Feng Liu and Yung-Fu Chang is dedicated to the Areca Nut, which is the fruit of the Areca palm (Areca catechu), which grows in most of the tropical Pacific Ocean (Melanesia and Micronesia). ), South Asia, Southeast Asia, and parts of eastern Africa. It is commonly referred to as betel nut. Consumption has many harmful health effects and is carcinogenic to humans. Various compounds present in nuts, including arecoline (the main psychoactive ingredient similar to nicotine), contribute to histological changes in the oral mucosa. It is known to be a major risk factor for cancer (squamous cell carcinoma) of the mouth and esophagus. However, in some cases, areca nut is used as a medicine for the treatment of certain diseases.

Many different review articles have been devoted to this plant, some of which were not reflected in the present review. Therefore, I suggest that the authors rework their article and explain to the reader how their article differs from those already published on the same topic.

For example

1) Arjungi K.N. Areca nut: a review. Arzneimittelforschung. 1976;26(5):951-6. PMID: 786304.

Journal of Public Health, Volume 45, Issue 1, March 2023, Pages 145–153, https://doi.org/10.1093/pubmed/fdab411

2) Manarangi De Silva, Leeanne Panisi, Fiona C. Brownfoot, Anthea Lindquist, Susan P. Walker, Stephen Tong, Roxanne Hastie Systematic review of areca (betel nut) use and adverse pregnancy outcomes Gynecology Obstetrics 2019 Vol 147, n 3 Pages 292 -300 https://doi.org/10.1002/ijgo.12971

3) Ye Jin Joo, David Newcombe, Vili Nosa & Chris Bullen (2020) Investigating Betel Nut Use, Antecedents and Consequences: A Review of Literature, Substance Use & Misuse, 55:9, 1422-1442, DOI: 10.1080/10826084.2019. 1666144

4) And others.

After revision, the review article can be further considered

Because my opinion about this article is MAJOR REVISION.

Sincerely

Reviewer

Many different review articles have been devoted to this plant, some of which were not reflected in the present review. Therefore, I suggest that the authors rework their article and explain to the reader how their article differs from those already published on the same topic.

For example

1) Arjungi K.N. Areca nut: a review. Arzneimittelforschung. 1976;26(5):951-6. PMID: 786304.

Journal of Public Health, Volume 45, Issue 1, March 2023, Pages 145–153, https://doi.org/10.1093/pubmed/fdab411

2) Manarangi De Silva, Leeanne Panisi, Fiona C. Brownfoot, Anthea Lindquist, Susan P. Walker, Stephen Tong, Roxanne Hastie Systematic review of areca (betel nut) use and adverse pregnancy outcomes Gynecology Obstetrics 2019 Vol 147, n 3 Pages 292 -300 https://doi.org/10.1002/ijgo.12971

3) Ye Jin Joo, David Newcombe, Vili Nosa & Chris Bullen (2020) Investigating Betel Nut Use, Antecedents and Consequences: A Review of Literature, Substance Use & Misuse, 55:9, 1422-1442, DOI: 10.1080/10826084.2019. 1666144

4) And others.

After revision, the review article can be further considered

So,  my opinion about this article is the MAJOR REVISION.

Author Response

The review article “The controversial roles of areca nut: medicine or toxin” by Pei-Feng Liu and Yung-Fu Chang is dedicated to the Areca Nut, which is the fruit of the Areca palm (Areca catechu), which grows in most of the tropical Pacific Ocean (Melanesia and Micronesia). ), South Asia, Southeast Asia, and parts of eastern Africa. It is commonly referred to as betel nut. Consumption has many harmful health effects and is carcinogenic to humans. Various compounds present in nuts, including arecoline (the main psychoactive ingredient similar to nicotine), contribute to histological changes in the oral mucosa. It is known to be a major risk factor for cancer (squamous cell carcinoma) of the mouth and esophagus. However, in some cases, areca nut is used as a medicine for the treatment of certain diseases.

Many different review articles have been devoted to this plant, some of which were not reflected in the present review. Therefore, I suggest that the authors rework their article and explain to the reader how their article differs from those already published on the same topic.

For example

1) Arjungi K.N. Areca nut: a review. Arzneimittelforschung. 1976;26(5):951-6. PMID: 786304.

Journal of Public Health, Volume 45, Issue 1, March 2023, Pages 145–153, https://doi.org/10.1093/pubmed/fdab411

 Ans: Thanks for the suggestion. We have included the article suggested by the reviewer in our revised manuscript as reference 9. We particularly focused on the literature of the recent 5 years. That is why we did not cite above article in the unrevised version of the MS.

2) Manarangi De Silva, Leeanne Panisi, Fiona C. Brownfoot, Anthea Lindquist, Susan P. Walker, Stephen Tong, Roxanne Hastie Systematic review of areca (betel nut) use and adverse pregnancy outcomes Gynecology Obstetrics 2019 Vol 147, n 3 Pages 292 -300 https://doi.org/10.1002/ijgo.12971

 Ans: We have included the article suggested by the reviewer in our revised manuscript as reference 110 and describing this article in lines 475-476. We collected the articles of AN but not betel nut. That is why we did not cite this article at the first time. Although AN also called betel nut, betel nuts are usually mixed with other materials, such as betel leaf, betel stem inflorescence and slaked lime. In order to exclude the effect caused by the materials other than AN, we did not describe the articles about betel nut mixed with materials. However, most of the cases in this suggested article use not only AN.

3) Ye Jin Joo, David Newcombe, Vili Nosa & Chris Bullen (2020) Investigating Betel Nut Use, Antecedents and Consequences: A Review of Literature, Substance Use & Misuse, 55:9, 1422-1442, DOI: 10.1080/10826084.2019. 1666144

Ans: We have included the article suggested by the reviewer in our manuscript as reference 103. We also included the description in our manuscript in lines 449-452. We collected the articles of AN but not betel nut. That is why we did not cite this article at the first time.  

4) And others.

Ans: We also have included another article, reference 8, suggested by the reviewer in our manuscript. We selected the recent 5 years articles and focused on AN but not on betel nut and updated recent trends of research in addition to acquiring new knowledge about AN. That is why we did not cite several articles in the unrevised version of the MS.

After revision, the review article can be further considered

Because my opinion about this article is MAJOR REVISION.

Reviewer 3 Report

In the manuscript entitled “The controversial roles of areca nut: medicine or toxin?” authors have summarized the individual chemical components of Areca nut (AN) and their various biological functions. They have emphasized both beneficial and toxic effects of AN and argued that removing or modifying the toxic ingredients of AN extractions may enhance their pharmacological activity for treating several diseases. The manuscript is written well, however more comprehensive details about the role of various components of AN can be added. All the tables need to be arranged carefully and aligned with target compound and their actions. It would be good if ‘several pharmacological effects’ in the effect category can be replaced with focused role mentioned in the reference paper.

The written english language in the manuscript is good.

Author Response

In the manuscript entitled The controversial roles of areca nut: medicine or toxin?” authors have summarized the individual chemical components of Areca nut (AN) and their various biological functions. They have emphasized both beneficial and toxic effects of AN and argued that removing or modifying the toxic ingredients of AN extractions may enhance their pharmacological activity for treating several diseases. The manuscript is written well, however more comprehensive details about the role of various components of AN can be added.

Ans: The effects of fatty acids were added in lines 170-188 as well as references 45-48. The biological materials were added in revised Table 1 which also contains the previous Table 2. The AN components, mechanisms and biological materials in experiments of AN extract, other mixture compounds and arecoline were described in the revised Tables 2 and 3, which represent the previous Tables 3 and 4. The role of these various components of AN will be clear in our revised MS.

All the tables need to be arranged carefully and aligned with target compounds and their actions.

Ans: The actions of each compound were organized in the revised Table 1, previous Table 2. The biological materials and references were added to this Table. This information will be helpful for the understanding the effects of each component.

It would be good if‘several pharmacological effects’in the effect category can be replaced with focused role mentioned in the reference paper.

Ans: The previous Table 1 was combined with previous Table 2. Therefore, the column‘several pharmacological effects’ does not exist anymore.

Reviewer 4 Report

In figure 1. The C, D, E, and F, should be more aesthetic, that is, the same size

Line 146: “acid, and dodecenoin acir” is correct as I describe it

In the tables of compounds, I recommend adding a column and placing in it the consecutive number that corresponds to each compound and those same numbers substitute them in the writing of the entire manuscript.

In table 1. I don't understand what N & M means

In the same table, the first line corresponding to the groups of compounds, these do not have compounds. I think that line should not go or, if applicable, place the name of the compound.

In addition to that it must describe what pharmacological effects they have and not "Several pharmacological effects"

On line 230, which stands for “BAX”

The first time that the scientific name of the bacteria is written, it is complete, and in the subsequent ones the genus is only written with the first literal and in capital letters, for example Escherichia coli, later E. coli.

Table 3 and 4. At the beginning of the sentence, correct in the "Pharmacological effects" column as well as mechanisms.

In Figure 2, I advise that you investigate the doses or concentrations that have the pharmacological and toxicological effects they suggest.

Figure 3. Corresponding to the human figure, there are programs that can better outline that human figure, and also to be better seen, I suggest placing a male and a female figure.

Author Response

In figure 1. The C, D, E, and F, should be more aesthetic, that is, the same size

Ans: Thanks for the suggestion. The size of figure 1. The C, D, E, and F were adjusted.

Line 146: “acid, and dodecenoin acir” is correct as I describe it

Ans: The ‘and ‘ was added in Line 168.

In the tables of compounds, I recommend adding a column and placing in it the consecutive number that corresponds to each compound and those same numbers substitute them in the writing of the entire manuscript.

Ans: The previous Table 1 was removed and merged with Table 2.

In table 1. I don't understand what N & M means

Ans: N & M mean calquiquelignan N and calquiquelignan M in ref 29. Two new flavonoids, calquiquelignan M (1), calquiquelignan N (2), along with nine known compounds (3–11), were isolated from the nuts of Areca catechu (Palmae). Their full name were described in the manuscript Line 122 -123.

In the same table, the first line corresponding to the groups of compounds, these do not have compounds. I think that line should not go or, if applicable, place the name of the compound. In addition to that it must describe what pharmacological effects they have and not "Several pharmacological effects"

Ans: The previous Table 1 was removed and merged into previous Table 2 as new Table 1.

On line 230, which stands for “BAX”

Ans: BAX is a apoptotic protein. We have added its full name in the manuscript in line 157.

The first time that the scientific name of the bacteria is written, it is complete, and in the subsequent ones the genus is only written with the first literal and in capital letters, for example Escherichia coli, later E. coli.

Ans: The abbreviations were corrected. 

Table 3 and 4. At the beginning of the sentence, correct in the "Pharmacological effects" column as well as mechanisms.

Ans: The words ‘and mechanisms’ were added in the titles of Tables 2 and 3 which represent the previous Tables 3 and 4.

In Figure 2, I advise that you investigate the doses or concentrations that have the pharmacological and toxicological effects they suggest.

Ans: The doses were added in the titles of Table 2 and 3 replacing the previous Tables 3 and 4 for describing pharmacological and toxicological effects.

Figure 3. Corresponding to the human figure, there are programs that can better outline that human figure, and also to be better seen, I suggest placing a male and a female figure.

Ans: Both male and a female have the same effects. Thus, we just showed male in the Figure 4, previous Figure 3. Corresponding to the human, which can simplify the meaning we would like to present.

Reviewer 5 Report

The manuscript presents a review with the nut of Areca cattechu Linn. (AN), a traditional medicine plant from Asia, with several psychoactive substances, commonly used such as tea or coffee, giving the users a lift sensation, but also with several unlike effects, such as oral cancer. Being used for both medicinal and recreational purposes, with several side effects, toxicological concerns have arisen. The authors begin the manuscript with the history of AN, following the chemical and biological properties.

Tables 1 and 3 are not necessary and should be excluded. Maybe Table 4 could also be excluded. The format of Figure 3 could be used to these tables.

Table 2 should be better explained/discussed.

The titles from 3.6, 3.7 and 4.9 should be changed.

Figures 2 and 3 should be better locate at the beggining of the manuscript.

Conclusions should be more focused on substances that confer the medicinal and also the toxicological effects. Also, the answer of the provocative title should be provided and better discussed.

Author Response

Tables 1 and 3 are not necessary and should be excluded. Maybe Table 4 could also be excluded. The format of Figure 3 could be used to these tables.

Ans: Thanks for the suggestion. The previous Table 1 was removed and merged into previous Table 2 as new Table 1. The previous Table 3 and Table 4, new Table 2 and Table 2, describe pharmacological and toxicological effects, respectively.

Table 2 should be better explained/discussed.

Ans: The previous Table 2, new Table 1, was extended to have columns of ‘Biological materials’ and ‘References’.

The titles from 3.6, 3.7 and 4.9 should be changed.

Ans: The titles were changed to ‘Skin protection‘ in line 324, ‘Anthelmintic activity‘ in line 340 and ‘Neuron activation‘ in line 498.

Figures 2 and 3 should be better locate at the beginning of the manuscript.

Ans: The previous Figure 2, new Figure 3, was moved to the end of the description of AN component to summarize the effects of AN components. The previous Figure 3, new Figure 4, was saved at the original place to summarize the pharmacological and toxicological effects of AN extract, other mixture compounds and arecoline.

Conclusions should be more focused on substances that confer the medicinal and also the toxicological effects. Also, the answer of the provocative title should be provided and better discussed.

Ans: The future trends and existing challenges of AN-related research were envisioned in the revised perspectives part in lines 562-585. Substances were focused on medicinal and toxicological effects referring to our title. Actually, AN components are a mixture of different components. Different components have different effects which are helpful or harmful for us. It is difficult to focus on specific substances. Therefore, to choose the appropriate substances for the researches’ purpose is very important. To chemically modify the structure of substances to fit our purpose might be a good option.

Reviewer 6 Report

1. I recommend extending the abstract - which results from the conducted review.

2. The article does not contain many important articles. Examples:

https://www.ncbi.nlm.nih.gov/pmc/articles/PMC4080659/

https://www.ncbi.nlm.nih.gov/pmc/articles/PMC8192186/

You should carefully review the literature and describe what is new in your review. The same applies to many subsections, e.g. fatty acids. The subsections are short and do not contain articles available in the literature. Example:

https://www.ncbi.nlm.nih.gov/pmc/articles/PMC8706666/

This applies to every section, not all available articles have been included. Sections should be more logically divided. They are too short at the moment. This is a review article, not a selection of articles. The literature should be reviewed carefully.

The article should be proofread.

Author Response

1. I recommend extending the abstract - which results from the conducted review.

Ans: Thanks for the suggestion. The abstract was extended by lines 15-21.

2. The article does not contain many important articles. Examples:

https://www.ncbi.nlm.nih.gov/pmc/articles/PMC4080659/

Garg, A.; Chaturvedi, P.; Gupta, P. C. A review of the systemic adverse effects of areca nut or betel nut. Indian J Med Paediatr Oncol 2014, 35, 3-9

Ans: We have included this article suggested by the reviewer in our manuscript as reference 8. We collected the articles of recent 5 years. That is why we did not cite this article in the unrevised version of the MS.

https://www.ncbi.nlm.nih.gov/pmc/articles/PMC8192186/

Athukorala, I. A.; Tilakaratne, W. M.; Jayasinghe, R. D. Areca Nut Chewing: Initiation, Addiction, and Harmful Effects Emphasizing the Barriers and Importance of Cessation. J Addict 2021, 2021, 9967097.

Ans: This article had already been included in the unrevised MS version as reference 100 and describing in lines 437-442.

You should carefully review the literature and describe what is new in your review. The same applies to many subsections, e.g. fatty acids. The subsections are short and do not contain articles available in the literature. Example:

https://www.ncbi.nlm.nih.gov/pmc/articles/PMC8706666/

Machová, M.; Bajer, T.; Šilha, D.; Ventura, K.; Bajerová, P. Volatiles Composition and Antimicrobial Activities of Areca Nut Extracts Obtained by Simultaneous Distillation-Extraction and Headspace Solid-Phase Microextraction. Molecules 2021, 26.

Ans: This article was cited as reference 77 in the revised version and describing in in lines 318-322. In addition, descriptions of fatty acids were added as references 45-48 and describing in in lines 170-188.

This applies to every section, not all available articles have been included. Sections should be more logically divided.

Ans: This article was logically divided four parts in lines 59-65. First part, section 1, describes the history of AN usage. The second part, section 2, introduces the components of AN and their biological effects. Arecoline is an important ingredient of AN in research and lots of articles focused on it. In addition, AN components are mixture of different components. AN extract has different effects from each component. And the effects of AN are controversial with pharmacological and toxicological effects. The third section introduces the pharmacological effects and the fourth sections explore the toxicological effects of AN extract, other mixture compounds and arecoline.

They are too short at the moment.

Ans: We added several paragraphs in this article. Lines 15-21, 50-65, 81-87, 170-188, 318-322, 449-452, 475-476, 508-513, 547-553, 562-585.

This is a review article, not a selection of articles. The literature should be reviewed carefully.

Ans: We selected the recent 5 years articles and focus on AN but not betel nut. We updated recent trend of research in addition to acquire new knowledge about AN. Moreover, this article focusses on the effect of AN (Figure 1C) but not betel nut (Figure 1D). Although AN also called betel nut, betel nuts are usually mixed with other materials, such as betel leaf, betel stem inflorescence and slaked lime. In order to exclude the effect caused by materials other than AN, we did not describe the articles about AN with the materials of betel nut as possible.

Reviewer 7 Report

As an unique and general comment, I am not sure that this type of review is really made for the Journal. This is long list of information but the authors do not try a critical analysis.

for me this article is taking place in a long list of review articles produced as cazzy that do not really impact the field, be not really useful.

Author Response

As an unique and general comment, I am not sure that this type of review is really made for the Journal. This is long list of information but the authors do not try a critical analysis.

For me this article is taking place in a long list of review articles produced as cazzy that do not really impact the field, be not really useful.

Ans: We selected the recent 5 years articles and focused on AN but not on betel nut and updated recent trends of research in addition to acquiring new knowledge about AN. We also provided perspectives and several trends.

Reviewer 8 Report

Areca nut (AN) chewing has often been attributed to increased risk of oral cancer with tumor formation but the AN's benefits have received much less attention, therefore it is important to understand both toxicological effects and potential pharmacological benefits of AN. This review offers an informative and educational summary of the scientific findings regarding AN. In general, the cited references are comprehensive and up to date, reflecting a careful and sufficient literature search. In Tables 1 and 3, it would be helpful to specify whether in vitro, in vivo, or both were used. In Table 2, references regarding each effect are missing. Section 5 is too general and too short. This is where the authors should have showed more effort to offer experts’ opinions with some in-depth discussions regarding the future trends and existing challenges of AN related research. Section 6 is just some redundant statements from section 1 (Introduction). Remove or revise this section. Figure 3 is not informative. Should include related components. Better to combine Figures 2 and 3.

Areca nut (AN) chewing has often been attributed to increased risk of oral cancer with tumor formation but the AN's benefits have received much less attention, therefore it is important to understand both toxicological effects and potential pharmacological benefits of AN. This review offers an informative and educational summary of the scientific findings regarding AN. In general, the cited references are comprehensive and up to date, reflecting a careful and sufficient literature search. In Tables 1 and 3, it would be helpful to specify whether in vitro, in vivo, or both were used. In Table 2, references regarding each effect are missing. Section 5 is too general and too short. This is where the authors should have showed more effort to offer experts’ opinions with some in-depth discussions regarding the future trends and existing challenges of AN related research. Section 6 is just some redundant statements from section 1 (Introduction). Remove or revise this section. Figure 3 is not informative. Should include related components. Better to combine Figures 2 and 3.

Author Response

Areca nut (AN) chewing has often been attributed to increased risk of oral cancer with tumor formation but the AN's benefits have received much less attention, therefore it is important to understand both toxicological effects and potential pharmacological benefits of AN. This review offers an informative and educational summary of the scientific findings regarding AN. In general, the cited references are comprehensive and up to date, reflecting a careful and sufficient literature search.

In Tables 1 and 3, it would be helpful to specify whether in vitro, in vivo, or both were used.

Ans: Thanks for the suggestion. The previous Table 1 was removed and merged into previous Table 2 as new Table 1. The previous Table 1 and Table 3, new Table 2, were added to the column of Biological materials.

In Table 2, references regarding each effect are missing.

Ans: A column of references was added to the previous Table 2 and new Table 1.

Section 5 is too general and too short. This is where the authors should have showed more effort to offer experts’ opinions with some in-depth discussions regarding the future trends and existing challenges of AN related research.

Ans: Section 6, previous section 5, was extended and the trends and challenges were described in lines 562-585.

Section 6 is just some redundant statements from section 1 (Introduction). Remove or revise this section.

Ans: Section 5, previous section 6, is conclusion. It briefly considers all of the sections in this article. We added several sentences throughout the lines 547-553.

Figure 3 is not informative. Should include related components. Better to combine Figures 2 and 3.

Ans: Previous Figure 3, new Figure 4, is used to summarize the effects of AN extract, other mixture compounds and arecoline. Previous Figure 2, new Figure 3, is used to summarize the effects of AN components.

Round 2

Reviewer 1 Report

NA

NA

Author Response

Comments and Suggestions for Authors

NA

Comments on the Quality of English Language

NA

Ans: Thanks for the suggestion.

Professor Hans-Uwe Dahms at Kaohsiung Medical University (Taiwan) assisted us in English correction. He taught English more than 8 years and ensures that the English in this paper is perfect (see the certificate).

Reviewer 2 Report

Pei-Feng Liu and Yung-Fu Chang's review article "Controversial Roles of the Areca Nut: Medicine or Toxin" focuses on the areca nut, which is the fruit of the areca palm (Areca catechu) native to most of the tropical Pacific Ocean (Melanesia and Micronesia). ), South Asia, Southeast Asia, and parts of eastern Africa. It is commonly referred to as betel nut. Consumption has many harmful health effects and is carcinogenic to humans. Various compounds present in nuts, including arecoline (the main psychoactive ingredient similar to nicotine), contribute to histological changes in the oral mucosa. It is known to be a major risk factor for cancer (squamous cell carcinoma) of the mouth and esophagus. However, in some cases, areca nut is used as a medicine for the treatment of certain diseases.

Many different publications are devoted to this plant. The presented article is interesting and has a certain value. The authors of the article took into account the comments of the reviewer, therefore the article can be published with minimal corrections. i.e, MINOR REVISION

Reviewer

The authors of the article took into account the comments of the reviewer, therefore the article can be published with minimal corrections in english grammatic. i.e, MINOR REVISION

Author Response

Comments on the Quality of English Language

The authors of the article took into account the comments of the reviewer, therefore the article can be published with minimal corrections in english grammatic. i.e, MINOR REVISION

Ans: Thanks for the suggestion.

Professor Hans-Uwe Dahms at Kaohsiung Medical University (Taiwan) assisted us with English correction. He taught English more than 8 years and ensures that the English in this paper is perfect (see the certificate).

Reviewer 4 Report

Refer to the program used and its characteristics, for the realization of the base structure. Place it in references.

Line 173: I don't understand what the “1” does in (200 µg/mL1)

Line 175, mg/kg is separated, this is correct, as it is found is as follows (mg/kg)

In table 1. (Vainillin, resveratrol, quinic acid, among others, please review the entire table) does not have specified biological material. It is also advisable to place the doses or concentrations used

800 µg/mLArecoline separate mL of Arecoline

In some tables in size and font is different, homogenize the entire manuscript

Figure 4, improve the human figure and place a feminine one as well

has no conclusions

Author Response

Comments and Suggestions for Authors

Refer to the program used and its characteristics, for the realization of the base structure. Place it in references.

Ans: We are very sorry that we don’t exactly understand what the meaning of ‘program’ is. If the program means the process of drawing pictures, the human shape in figure 4 is our hand-drawing using Inkscape (https://inkscape.org/zh-hant/). All of the other figures were generated by PPT. The sentence of ‘This figure is our hand-drawing using Inkscape (https://inkscape.org/zh-hant/).’ as added in lines 511-512.

Line 173: I don't understand what the “1” does in (200 µg/mL1)

Ans: Thanks for the suggestion. The ‘1’ was deleted.

Line 175, mg/kg is separated, this is correct, as it is found is as follows (mg/kg)

Ans: The ‘mg/ kg’ was corrected as ‘mg/kg’.

In table 1. (Vainillin, resveratrol, quinic acid, among others, please review the entire table) does not have specified biological material. It is also advisable to place the doses or concentrations used

Ans:

The biological materials were added.

Because AN is a mixture of several compounds, the amount of each component in AN may not be high enough to cause their effects. We describe the possible effects of each component in this article but not noting their doses nor concentrations. Therefore, the effective doses or concentrations were not included in table 1.

800 µg/mL Arecoline separate mL of Arecoline

Ans: The µg/mL and Arecoline in Table 3 (ref 94) were separated.

In some tables sizes and fonts are different, homogenize the entire manuscript

Ans: The size and font in all of the tables were adjusted consistently throughout the manuscript.

Figure 4, improve the human figure and place a feminine one as well

Ans: The figure was re-drawn. The sentences ‘Both male and a female have the same effects, except reproduction. This figure doesn’t indicate any gender’ was inserted in lines 510-511. The effects especially works on female, such as ‘Reproduction’ and ‘Abortifacient effects’ were added with sexual symbol. The sexual symbols of both gender were also added to this figure.

Has no conclusions

Ans: Conclusions were moved to section 6 instead of previous section 5.

Reviewer 6 Report

The authors took into account the recommended advice.

A native speaker review is recommended.

Author Response

Comments and Suggestions for Authors

The authors took into account the recommended advice.

Comments on the Quality of English Language

A native speaker review is recommended.

Ans: Thanks for the suggestion.

Professor Hans-Uwe Dahms at Kaohsiung Medical University (Taiwan) assisted us in English correction. He taught English more than 8 years and ensures that the English in this paper is perfect (see the certificate).

Reviewer 7 Report

I stayed on my first opinion and find that this manuscript do not relates to the Journal.

The authors have to contact a journal devoided to natural products

Author Response

Comments and Suggestions for Authors

I stayed on my first opinion and find that this manuscript do not relates to the Journal.

The authors have to contact a journal devoided to natural products

Ans:

This journal ‘International Journal of Molecular Sciences (IJMS)’ contains the section of ‘bioactives’. And ‘Antitumor/Anti-inflammatory Activities of Natural Compounds from Plants 3.0’ is one of a special issue. The compounds of AN are natural compounds from plants with bioactivity. We also described the molecular mechanism of AN extracts and components. We believe our article is qualified to be considered in this journal.

Round 3

Reviewer 4 Report

No comment

Reviewer 7 Report

As I said before, this article is not of quality enough  to be published by this Journal. This is like a catalog and do not give any new informations.

So, Reject